# Emissions of Volatile Organic Compounds (VOCs) from an Open-Circuit Dry Cleaning Machine Using a Petroleum-Based Organic Solvent: Implications for Impacts on Air Quality

**Hyeonji Lee [1,†], Kyunghoon Kim [2,†], Yelim Choi [2] and Daekeun Kim [2,*]**

[1] National Air Emission Inventory and Research Center, 206 Osongsaengmyeong-ro, Osong-eup, Cheongju 28166, Korea; dlswjfal@korea.kr

[2] Department of Environmental Engineering, Seoul National University of Science and Technology, 232 Gongneung-ro, Nowon-gu, Seoul 01811, Korea; kyunghoon.kim@seoultech.ac.kr (K.K.); yelim9502@seoultech.ac.kr (Y.C.)

[*] Correspondence: kimd@seoultech.ac.kr; Tel.: +82-2-970-6606

[†] These authors contributed equally to this work.

**Abstract:** Volatile organic compounds (VOCs) are known to play an important role in tropospheric chemistry, contributing to ozone and secondary organic aerosol (SOA) generation. Laundry facilities, using petroleum-based organic solvents, are one of the sources of VOCs emissions. However, little is known about the significance of VOCs, emitted from laundry facilities, in the ozone and SOA generation. In this study, we characterized VOCs emission from a dry-cleaning process using petroleum-based organic solvents. We also assessed the impact of the VOCs on air quality by using photochemical ozone creation potential and secondary organic aerosol potential. Among 94 targeted compounds including toxic organic air pollutants and ozone precursors, 36 compounds were identified in the exhaust gas from a drying machine. The mass emitted from one cycle of drying operation (40 min) was the highest in decane (2.04 g/dry cleaning). Decane, nonane, and *n*-undecane were the three main contributors to ozone generation (more than 70% of the total generation). *N*-undecane, decane, and *n*-dodecane were the three main contributors to the SOA generation (more than 80% of the total generation). These results help to understand VOCs emission from laundry facilities and impacts on air quality.

**Keywords:** volatile organic compounds; emissions; open-circuit drying machine; air quality

## 1. Introduction

Volatile organic compounds (VOCs) are organic chemicals with boiling points less than or equal to 250 °C and originate from natural or anthropogenic sources [1]. The anthropogenic sources emitting VOCs refer to human activities, including vehicle emissions, petrochemical industry emissions, residential combustion, and solvent use [2,3]. VOCs emitted from the various sources play a crucial role in tropospheric chemistry. Specifically, parts of the emitted VOCs, that are highly reactive, are known as major precursors to secondary air pollutants, such as ozone ($O_3$) and secondary organic aerosol (SOA) through photochemical reactions [4–6].

Among emission sources of VOCs, volatile chemical products containing organic solvents are emerging as the largest emission source in urban area in 2010s. For example, VOCs emission sources in the United States are mostly composed of consumer solvents use, industrial solvents use, road transport, and industrial processes [7]. Among these emission sources, solvents use accounted for more than 50% of total VOCs emission in the United States. In addition, a previous report on air pollutants emission in the Republic of Korea revealed that the nation emitted about 1,024,029 metric tons of anthropogenic VOCs in 2016, with the solvents use category accounting for the largest portion of the emissions

(54%) [8]. Facilities using organic solvents include painting facilities, print shops, laundry facilities, and other organic solvent use facilities.

Characteristics of VOCs emission (e.g., composition of VOCs emitted, proximity to residential area) are different depending on the facilities using organic solvents. For example, the composition of VOCs emitted can vary with different facilities because they use different volatile chemical products, according to their technical needs. The effect of the emitted VOCs on air quality can be different with the composition of the VOCs, because each VOC has different abilities to form ozone or SOA per unit weight of VOC [9,10]. Moreover, the facilities have different distances from residential areas; print shop and laundry facilities are likely to be close to residential areas while painting facilities in industry and wastewater treatment plants are far from these areas. Difference in proximity to residential area among the emission sources may cause a difference in the effects on air quality, resulting from chemical reactions in the atmosphere between VOCs and other substances (e.g., NOx, SOx), because compositions of NOx and SOx are different with location [11,12].

Among facilities using organic solvents, laundry facilities are equipped with dry-cleaning machines, where various VOCs are used as components of chemical solvents. Two types of dry-cleaning machines are used in laundry facilities: closed-circuit and open-circuit machines. Closed-circuit machines have an enclosed structure, where solvent is condensed by the drying air inside the machines, without general venting systems [13,14]. On the other hand, open-circuit machines involve two separate processes: a cleaning process along with a drying process using hot air vented to atmosphere. For open-circuit machines, VOCs can be emitted during the drying process. In addition, laundry facilities are generally located around residential areas while other emission sources of VOCs (e.g., industry, wastewater treatment, biomass burning) are far from those areas. Considering the site-specific characteristics of VOCs emission sources and their effects on air quality, further studies are needed to investigate the emission characteristics of laundry facilities and the effects of VOCs emission from laundry facilities on air quality. However, research has not been conducted to investigate the contribution of VOCs, emitted by laundry facilities, to the generation of ozone and secondary organic aerosol (SOA).

This study has been conducted to characterize VOCs emission from a small-scale dry-cleaning process, operated with the open-circuit machine using petroleum organic solvents. Qualification and quantification of the exhaust gases have been performed to assess the impact of the dry-cleaning VOCs on air quality by using photochemical ozone creation potential (POCP) and secondary organic aerosol potential (SOAP).

## 2. Materials and Methods

### 2.1. Dry Cleaning Process

An at-laundry dry cleaning machine includes washing machines, dryers, and electric steam boilers [15]. Washing machines remove contaminants and stains on textile products by using organic solvents, and dryers evaporate the remnants of solvent on textile products which result from the washing process, with an electric steam boiler supplying hot air into the dryer.

In this study, we simulated a laundry dry cleaning process in our laboratory by installing dry cleaning machines that are actually used in households in Korea (Figure S1). We used a 13 kg washing machine (ESE-7313, Eunsung Engineering, Daegu, Korea), a 15 kg dryer (SR-7615, Eunsung Engineering, Daegu, Korea), and an 8 kg/h electric steam boiler (PHE-5, Pyeonghwa-boiler, Daegu, Korea) in our experiment.

In the washing and drying processes, 3 kg of 100% cotton fiber was used as laundry. The washing process took 23 min: (1) main washing (10 min) after the solvent in the solvent tank was refueled in the washing machine containing the laundry textile, (2) rinse washing (5 min), and (3) deoiling process (8 min). The drying process used hot air supplied from the electric steam boiler, and was operated at 40 °C in 40 min. The air temperature in the laboratory was kept at 25 °C throughout the whole process.

To examine the effects of operational conditions on the concentration of total volatile organic compounds (TVOCs) in exhaust gas, we changed the operating temperature (i.e., 40, 50, 60 °C) and time (i.e., 30, 40, 50 min) in dry unit, weight of laundry (i.e., 3, 6, 10 kg), and material of laundry clothes (i.e., cotton, polyester, wool, silk).

### 2.2. VOC Sampling and Analysis

VOCs were qualified and quantified by using Gas chromatography-mass spectrometry (GC-MS). The gas was sampled at five different time points (0, 4, 8, 12, and 18 min of dry-cleaning process), through low-bleed and high-puncture-tolerant silicone gas chromatography septa installed at the sampling ports of a sampling chamber connected to a gas vent at the back of the dryer. The sampling chamber was made of acrylic with a total volume of 30 L (length 54 cm, diameter 26 cm, thickness 1 cm). A 1.5 cm (internal diameter) sampling port was installed in the chamber in the direction of exhaust gas flow to directly collect dry cleaning gas. The gas was collected for 30 s at a flow rate of 0.1 L/min by connecting a solid sorbent tube, filled with Tenax TA (APK Sorbent Tube, KNR Co., Ltd., Namyangju, Gyeonggi-do, Korea) as an adsorbent, to a suction pump controlling the gas flow rate. We analyzed the exhaust gas as soon as the sampling was completed. Otherwise, the sorbent tubes were stored at 4 °C in the case of delayed analysis. Analyses of the gas samples began by desorbing the exhaust gas from the solid sorbent, using low temperature thermal desorption equipment (UNITY-XR, MARKES International, Sacramento, California, USA) connected to a thermal desorber-gas chromatography/mass spectrometry (GC/MS: 7820A/5977B, Agilent Technologies, USA; GC column: DB-1, 60 m × 0.32 mm × 3 µm, Agilent Technologies, Santa Clara, California, USA). The gas samples were desorbed for 10 min at 300 °C with the low temperature thermal desorption equipment. The initial temperature of the GC oven was 50 °C, held for 10 min, then increased to 220 °C (heating rate, 5 °C/min), and held for another 10 min. Helium (99.999%) was used as the carrier gas at a flow rate of 1.5 mL/min. A total of 94 substances, including Toxic Organics-15 standard gas (Restek, Bellefonte, Pennsylvania, USA) and ozone($O_3$) precursor standard gas (Supelco, Bellefonte, Pennsylvania, USA), were used for qualification and quantification of the VOCs. The dry-cleaning operation was triplicated and all VOCs analyses were conducted. We presented average concentrations in this paper.

The concentrations of total volatile organic compounds (TVOC) were determined by summing all detected peaks of gas chromatography-flame ionization detector (GC-FID) from *n*-hexane (C6) to *n*-hexadecane (C16), and expressed as toluene equivalent. Gas-phase samples for TVOC analysis were taken every 2 min over 40 min of dry-cleaning operation, with 0.5 mL gas-tight syringes at the sampling ports of the sampling chamber. The concentration of TVOC was analyzed using gas chromatography (YL 6500 GC, YL Instruments Co., Ltd., Korea) equipped with a flame ionization detector (FID) and a capillary HP-5 (30 m × 0.32 mm i.d., 0.1 µm film thickness). The split (1:5) injection mode with the volume of 300 mL and an injector temperature of 200 °C was maintained. The GC oven temperature was programmed at 230 °C for 2 min. The carrier gas (He, 99.999% purity) flow rate was set at 3 mL/min. The FID detector was used with a fuel gas flow (hydrogen) of 30 mL/min, and an oxidizing gas flow (air) of 300 mL/min. The detector temperature was set to 250 °C.

### 2.3. Statistical Analysis

To predict the concentrations of VOCs at the exhaust gas from the first 18 min to the end of dry-cleaning process (40 min), we employed the five-parameter Weibull function using the Semi-Log Scatter Plot of Sigma Plot 12.5. Specifically, the concentration area of each VOC over time was calculated to characterize the temporal change in the concentrations of individual VOCs emitted throughout the drying process.

We also conducted the principal component analysis (PCA) to classify each VOC depending on the time point at which the maximum concentration appeared during the drying process. For the PCA, we used the 'pheatmap' function in R studio version 1.3.1073.

We also generated a heat map showing relative concentrations of VOCs with different time points (i.e., 0, 4, 8, 12, 18 min). A dendrogram in the heat map showed the clustering of the relative concentrations based on the similarity of VOCs occurrence with time. The relative concentrations of each compound with different time points was graded (from −1.5–1.5) using color code in the heat map.

### 2.4. Estimation of POCP and SOAP

2.4.1. Estimation of POCP

VOCs are regarded as main ozone precursors that photochemically react with nitrogen oxides to produce ozone [16]. Ozone production depends on the concentration ratio of nitrogen oxides and VOCs and reactivity of each VOC [17]. The POCP represents the relative ozone formation potential (OFP) of specific species, with the POCP value of ethylene set as 100 (a reference value). The POCP values for VOCs have been well studied by investigating the ozone formation mechanism involving VOCs [9,10]. Since the POCP values are determined by running a photochemical trajectory model in a specific region, they can be changed according to the weather and spatial characteristics of the region. To date, simulations for most POCP values have been focused on Europe. To characterize the significance of the emission source (dryers) in ozone generation, the $POCP_{\text{weighted emission}}$ (μg/dry cleaning) was calculated using the POCP values, suggested by Derwent in 2007 [18]. The $POCP_{\text{weighted emission}}$ for species, $i$, was evaluated as follows.

$$POCP_{\text{weighted emission}} = E_i \times POCP_{\text{species,i}} \tag{1}$$

where, $E_i$ represents the emission (μg/dry cleaning) of VOC species, $i$, from one-time dry-cleaning operation.

2.4.2. Estimation of SOAP

VOCs originating from anthropogenic and biologic sources are known to generate SOA by oxidation reaction with OH radicals, $NO_3$ radicals and $O_3$ in the atmosphere. Understanding of VOCs as precursors to SOA is significant in controlling the formation of PM 2.5. SOAP represents the propensity of each organic compound to form SOA, with the SOAP value of toluene set as 100 (a reference value). To quantify the significance of the emission source (dryers) in SOA generation, $SOAP_{\text{weighted emission}}$ (μg/dry cleaning) with species, $i$, was calculated using the SOAP values, developed by Derwent [19].

$$SOAP_{\text{weighted emission}} = E_i \times SOAP_{\text{species,i}} \tag{2}$$

## 3. Results

### 3.1. Concentrations of VOCs during Dry Cleaning Process

Among 94 standard materials, 36 compounds were quantified above the method detection limit (MDL) (Table S1, Figure 1). The highest concentration was observed in hexane at 12 min (13,055 ppb), followed by methylene chloride (10,626 ppb) and acetone (10,595 ppb). Previous studies have reported the characteristics of VOCs generated from small dry cleaners, and found that the major substances detected included toluene, benzene, xylene, styrene and ethylbenzene [10,18,19], being consistent with our results. In Figure 1, more than a half of the compounds had their maximum concentrations around 12 min and only a few compounds (e.g., ethyl acetate, butane, vinyl acetate) had them at the beginning of the drying process (0–4 min).

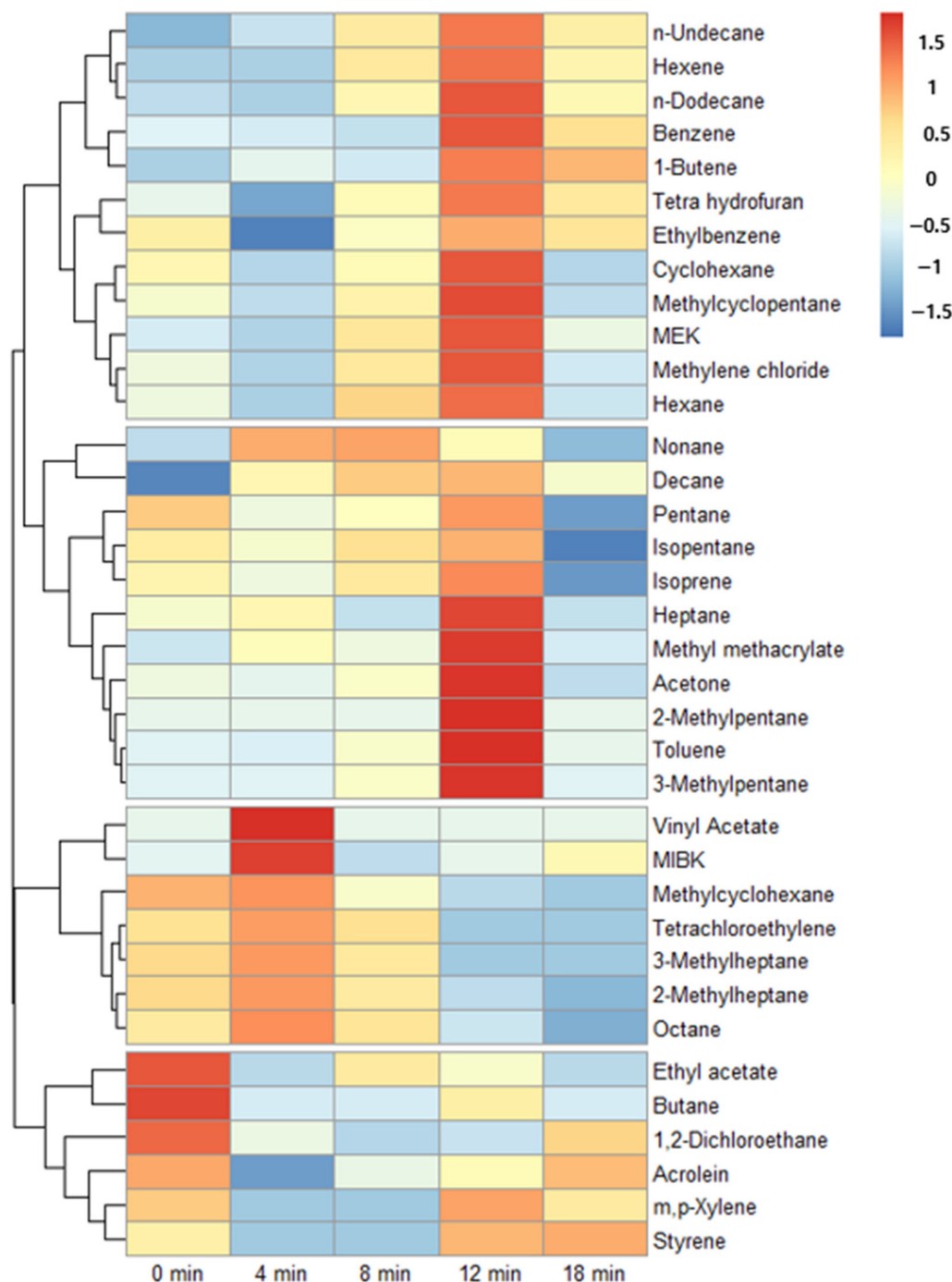

**Figure 1.** Heat map of relative VOCs concentrations over the first 18 min of drying process. Principal component analysis was performed using R studio (version 1.3.1073) for aggregating analytes into different groups depending on the time point at which maximum concentration appeared. The relative concentrations are on the natural logarithmic scale.

The concentration of total 36 VOCs increased from the beginning to the first 12 min and then decreased (Table S1); 5785 ppb (0 min), 11,342 ppb (4 min), 15,362 ppb (8 min), 49,834 ppb (12 min), and 7055 ppb (18 min). *N*-dodecane, *n*-undecane, decane, and nonane, which are compounds with the four lowest vapor pressures (0.236–4.96 mmHg), had relatively consistent concentrations throughout the experiment. On the other hand, compounds with relatively high vapor pressures (e.g., cyclohexane, methylcyclopentane, hexane, acetone, methylene chloride) were observed to have increasing trends in their concentrations from 0 min to 12 min and then decrease afterward, indicating that compounds with relatively

low vapor pressures might be mostly volatilized into gas phase within the drying period (40 min) instead of remaining on the laundry surface.

In addition to the chemical properties of compounds, some operational conditions also influenced the TVOC concentration in exhaust gas (Figure 2). The operating temperature in the dry unit was associated with the TVOCs concentration; the lower operating temperature (40 °C) resulted in the lower TVOCs concentration. The weight of laundry was proportional to the TVOCs concentration. Moreover, TVOCs were observed to be emitted mostly in the beginning of drying process for 3 kg of laundry weight while they were emitted throughout the entire operating period (40 min) for both 6 kg and 10 kg of laundry weight. TVOCs were slowly emitted from laundry made of silk, compared to laundry made of other materials (i.e., cotton, polyester, wool).

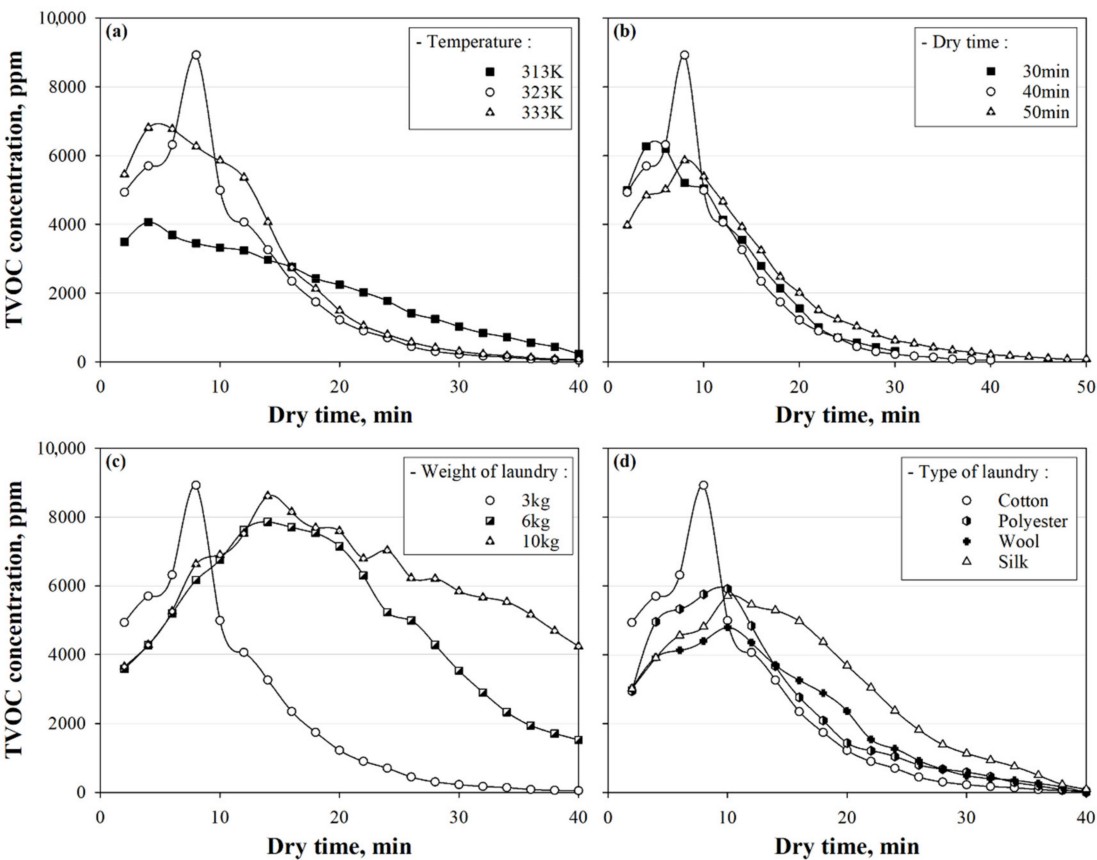

**Figure 2.** The concentrations of TVOCs in exhaust gas during drying processes with different operational conditions: (**a**) operating temperature in dry unit; (**b**) operating time in dry unit; (**c**) weight of laundry; (**d**); type of laundry cloth. For the operating condition in the dry-cleaning process, defaults were applied, except for the variable parameters. Default conditions were 3 kg of laundry weight (cotton); 40 min of drying time; 40 °C of operating temperature. Abbreviation: total volatile organic compound (TVOC).

### 3.2. Emission Factors for VOC Species

Mass of each compound, emitted from one cycle of drying operation (40 min in this study), was presented in Table 1. The highest mass generated from one cycle of operation was observed in decane (2,035,712 µg/dry cleaning), followed by nonane (1,362,063 µg/dry cleaning), *n*-undecane (1,292,820 µg/dry cleaning), hexane (1,093,942 µg/dry cleaning), and 1,2-dichloroethane (827,240 µg/dry cleaning); these top five VOC species accounted for more than 75 % of the total amount of VOC emission. Although decane was the compound with the second highest composition (34.42%) in the organic solvent (Table S2), its mass emitted from one cycle of drying operation was the highest among the compounds. The

mass of emitted *n*-undecane was about a half of that of decane, even though *n*-undecane was the compound with the highest composition (43.11%) in the organic solvent.

**Table 1.** Mass of VOCs emitted (µg/dry cleaning) from one cycle of drying operation.

|      | Compounds | Mass Generated |      | Compounds | Mass Generated |
| ---- | --------- | -------------- | ---- | --------- | -------------- |
| (1)  | Decane | 2,035,713 | (19) | Acrolein | 15,349 |
| (2)  | Nonane | 1,362,064 | (20) | 2-Methylpentane | 7512 |
| (3)  | *n*-Undecane | 1,292,821 | (21) | Methyl ethyl ketone | 6191 |
| (4)  | Hexane | 1,093,942 | (22) | Benzene | 4835 |
| (5)  | 1,2-Dichloroethane | 827,240 | (23) | Ethylbenzene | 4733 |
| (6)  | Methylene chloride | 709,959 | (24) | Tetrachloroethylene | 4446 |
| (7)  | Acetone | 627,845 | (25) | Heptane | 3702 |
| (8)  | Octane | 258,671 | (26) | Pentane | 3311 |
| (9)  | *n*-Dodecane | 126,620 | (27) | 1-Butene | 2614 |
| (10) | Methylcyclopentane | 92,477 | (28) | Methyl methacrylate | 2241 |
| (11) | Cyclohexane | 86,727 | (29) | Styrene | 1959 |
| (12) | 2-Methylheptane | 37,636 | (30) | Methyl isobutyl ketone | 1712 |
| (13) | 3-Methylheptane | 29,288 | (31) | *m, p*-Xylene | 1649 |
| (14) | Toluene | 25,930 | (32) | Ethyl acetate | 748 |
| (15) | Isopentane | 24,410 | (33) | Butane | 743 |
| (16) | 3-Methylpentane | 16,969 | (34) | Vinyl Acetate | 726 |
| (17) | Tetra hydrofuran | 16,345 | (35) | Isoprene | 632 |
| (18) | Methylcyclohexane | 15,493 | (36) | Hexene | 250 |

*3.3. Photochemical Ozone Creation Potential (POCP) of VOCs*

POCP$_{\text{weighted emission}}$ of VOCs discharged during drying processes was estimated for 25 compounds whose POCP values are available in the previous literature (Figure 3). The total POCP$_{\text{weighted emission}}$ of 25 compounds was estimated as 282.4 POCP$_{\text{weighted emission}}$ (data not shown). Among the 25 compounds, Table 2 shows the top 10 VOCs that contributed to $O_3$ generation. The top 10 VOCs accounted for more than 98% of the total $O_3$ generation. Of the 25 ozone precursors included in the POCP calculation, decane (91.6 POCP$_{\text{weighted emission}}$), nonane (61.3 POCP$_{\text{weighted emission}}$) and n-undecane (51.7 POCP$_{\text{weighted emission}}$) were the three main contributors to $O_3$ generation. Decane, with the highest mass generated from one cycle of operation (Table 1), had the highest contribution to ozone production as well, accounting for 32.44% of total POCP$_{\text{weighted emission}}$.

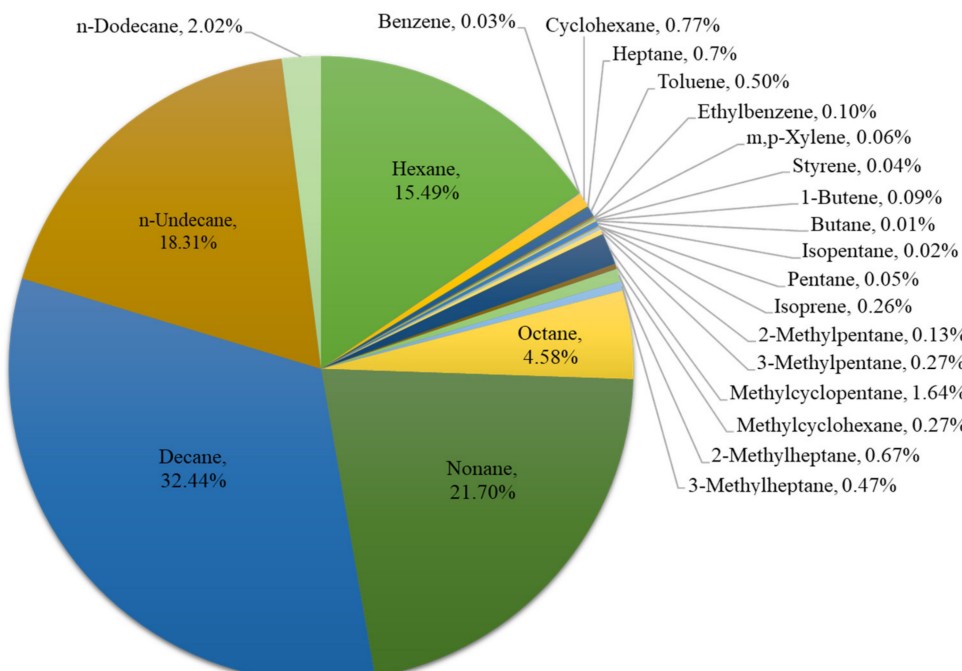

**Figure 3.** POCP$_{\text{weighted emission}}$ contributions (%) of dry-cleaning VOCs emitted from small-scale dry-cleaning operation.

**Table 2.** Top 10 VOCs that contributed to O$_3$ generation and their POCP$_{\text{weighted emission}}$ (g/dry cleaning).

|  | Compounds | POCP-Value | POCP$_{\text{weighted-emission}}$ |
|---|---|---|---|
| (1) | Decane | 45 | 91.6 |
| (2) | Nonane | 45 | 61.3 |
| (3) | *n*-Undecane | 40 | 51.7 |
| (4) | Hexane | 40 | 43.8 |
| (5) | Octane | 50 | 12.9 |
| (6) | *n*-Dodecane | 45 | 5.7 |
| (7) | Methylcyclopentane | 50 | 4.6 |
| (8) | Cyclohexane | 25 | 2.2 |
| (9) | 2-Methylheptane | 50 | 1.9 |
| (10) | Toluene | 55 | 1.4 |

*3.4. Second Organic Aerosol Potential (SOAP) of VOCs*

Among our studied compounds, only 28 VOCs were included in the SOAP estimation (Figure 4). The total SOAP$_{\text{weighted emission}}$ of 28 compounds was estimated as 46.7 SOAP$_{\text{weighted emission}}$ (data not shown). Among the 28 compounds, Table 3 shows the top 10 VOCs with the highest SOAP$_{\text{weighted emission}}$ during the drying process. The top 10 VOCs accounted for more than 99% of the total SOAP$_{\text{weighted emission}}$. *N*-undecane, who had the third highest total mass generated from one cycle of operation (Table 1), had the highest SOAP$_{\text{weighted emission}}$ accounting for 44.78% of the total SOAP$_{\text{weighted emission}}$. The second highest SOAP$_{\text{weighted emission}}$ was observed in decane with the highest total mass generated from one cycle of operation, accounting for 30.44% of the total SOAP$_{\text{weighted emission}}$. VOCs that were emitted with relatively high mass during the dry-cleaning process had higher SOAP$_{\text{weighted emission}}$ as well.

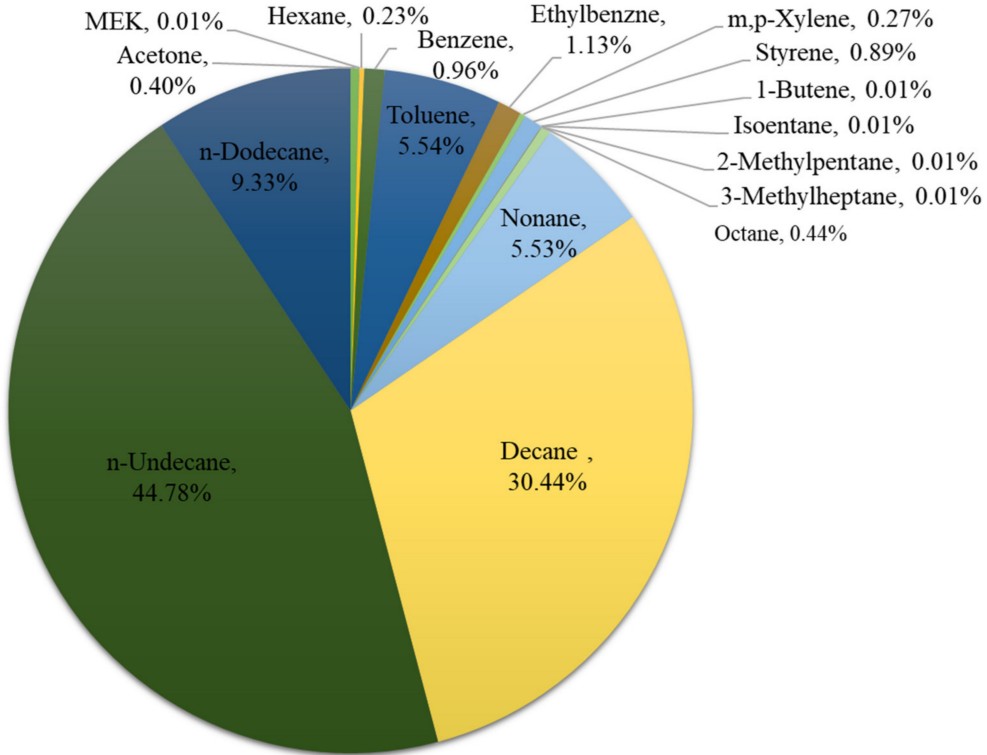

**Figure 4.** SOAP$_{\text{weighted emission}}$ contributions (%) of dry-cleaning VOCs emitted from small-scale dry-cleaning operation.

**Table 3.** Top 10 VOCs that contributed to SOA generation and their SOAP$_{\text{weighted emission}}$ (g/dry cleaning).

|  | Compounds | SOAP-Value | SOAP$_{\text{weighted-emission}}$ |
|---|---|---|---|
| (1) | *n*-Undecane | 16.2 | 20.9 |
| (2) | Decane | 7 | 14.2 |
| (3) | *n*-Dodecane | 34.5 | 4.4 |
| (4) | Toluene | 100 | 2.6 |
| (5) | Nonane | 1.9 | 2.6 |
| (6) | Ethylbenzene | 111.6 | 0.5 |
| (7) | Benzene | 92.9 | 0.4 |
| (8) | Styrene | 212.3 | 0.4 |
| (9) | Octane | 0.8 | 0.2 |
| (10) | Acetone | 0.3 | 0.2 |

## 4. Discussion

VOCs have diverse anthropogenic emission sources with different emission characteristics (e.g., composition of emitted VOCs, proximity to residential area). Among the VOCs emission sources, VOCs emission from laundry facilities and its impact on air quality were mostly unexplored. In the current study, we simulated the dry-cleaning operation of laundry facilities to characterize the VOCs emission during the operation with time frame and further quantitatively assessed the impact of the emitted VOCs from laundry facilities on air quality using POCP and SOAP. A total of 36 compounds were detected above the MDL, and hexane presented the highest concentration among the detected compounds. Decane and n-undecane were observed to be the two dominant compounds in both ozone and SOA generation.

We observed that the characteristics of VOCs emission from laundry facilities were different with chemical properties of compounds. Overall, we observed that the highest concentrations in exhaust gas of the drying unit appeared within the first 12 min for most

compounds (Figure 1) and the concentrations decreased after this. However, for less volatile compounds (e.g., n-dodecane, n-undecane, decane, nonane), their concentrations in exhaust gas were relatively less variant throughout the entire drying process, indicating that chemical properties (i.e., vapor pressure) can affect the emission pattern with time. In addition, we found that vapor the pressure of each compound can affect the mass emitted from one cycle of drying operation (Table 1 and Table S2). The mass of decane emitted was the highest, although its composition in the organic solvent was the second highest. The relatively high vapor pressure of decane (1.73 mmHg), compared to that of u-undecane (0.629 mmHg), may account for this result. These results may help to understand the emission characteristics of diverse VOCs with time frame and further design of the adsorber of VOCs emitted from laundry facilities, considering the emission characteristics and chemical properties of each VOC.

Among the 36 detected compounds in this study, we found a couple of dominant compounds affecting ozone and SOA generation, such as hexane, nonane, decane, n-undecane, and *n*-dodecane. Their contributions were higher than 80% of the total amount of generated ozone and SOA. Thus, it may be recommended for organic solvent manufacturers to remove those compounds from their products for reducing the amount of ozone and SOA generation from laundry VOCs.

The results in our study allow us to understand the characteristics of the VOCs emission during the drying process, specifically using an open-circuit drying machine with a petroleum-based organic solvent and reveal the contribution of the emitted VOCs to ozone and SOA generation. From a local perspective, our study may also help to establish atmosphere-related policies applicable to laundry facilities in Korea.

## 5. Conclusions

This study investigated VOC emissions during the drying processes of a small laundry establishment and the contribution of the emitted VOCs to ozone and SOA generation. VOCs were mostly emitted from the drying processes at the beginning of the drying process, and a small amount was released after a certain period of time. Among our studied compounds, decane, nonane, and *n*-undecane were the compounds with the highest mass emitted during the drying process. Decane and *n*-undecane were observed to be the biggest contributors to both ozone and SOA generation. To improve atmospheric air quality from VOCs emission in laundry facilities, further equipment may be recommended to adsorb the dominant compounds (e.g., decane, *n*-undecane) in the ozone and SOA generation.

**Supplementary Materials:** The following are available online at https://www.mdpi.com/article/10.3390/atmos12050637/s1, Table S1: Volatile organic compounds detected in dry cleaning gas and their concentrations during the first 18 min of drying operation, Table S2: Composition of VOCs in the organic solvent used in this study, Figure S1: Schematic drawing of dry-cleaning process employed in this study.

**Author Contributions:** Conceptualization, D.K.; methodology, H.L. and D.K.; formal analysis, H.L. and K.K.; investigation, H.L. and Y.C.; resource, D.K.; data curation, H.L. and Y.C.; writing—original draft preparation, H.L. and K.K.; writing—review and editing, K.K. and D.K.; visualization, H.L. and Y.C.; supervision, D.K.; project administration, D.K.; funding acquisition, D.K.; All authors have read and agreed to the published version of the manuscript.

**Funding:** This work was supported by the Technology Development Program to Solve Climate Changes of the National Research Foundation (NRF) funded by the Ministry of Science, ICT (2017M1A2A2086647) and Basic Science Research Program through the National Research Foundation of Korea (NRF) funded by the Ministry of Education (2020R1A6A1A03042742). K.K. acknowledges the financial support from Seoul Nation University of Science and Technology.

**Institutional Review Board Statement:** Not applicable.

**Informed Consent Statement:** Not applicable.

**Data Availability Statement:** The data presented in this study are available on request from the corresponding author.

**Acknowledgments:** We acknowledge the technical assistance of Green Environmental Complex Center.

**Conflicts of Interest:** The authors declare no conflict of interest. The funders had no role in the design of the study; in the collection, analyses, or interpretation of data; in the writing of the manuscript; or in the decision to publish the results.

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
