# Peer review of "Emissions of Volatile Organic Compounds (VOCs) from an Open-Circuit Dry Cleaning Machine Using a Petroleum-Based Organic Solvent: Implications for Impacts on Air Quality"

_atmosphere, doi:10.3390/atmos12050637_

Round 1

Reviewer 1 Report

This study investigated the VOC emissions during drying processes of a small dry cleaning laundry machine. The effective contributions of the emitted VOCs to ozone and SOA generations are estimated with the reported photochemical ozone creation potential (POCP) and secondary organic aerosol potential (SOAP). While the results of this work seem interesting in some aspects (providing interesting data about the VOC concentrations), the scholarly level can be greatly improved. For example,

(1) The authors did not explain why a "heat map" is plotted for the normalized VOC concentrations (Figure 1) and what is the meaning of Figure 1. How did they normalize the VOC concentrations? 

(2) The authors did not explain why the five-parameter Weibull function was used and the meaning of such analysis. 

(3) The POCP and SOAP are from model simulation works that were performed more than 10 years ago. The authors did not discuss how good or how bad these modeled results are. 

(4) The title only mention "dry machine". However, I feel "dry cleaning" should also be mentioned. 

(5) There is no page number in the Supporting Material. 

(6) The authors claimed "Thus, it may be recommended for organic solvent manufacturers to remove those compounds from their products for reducing the amount of ozone and SOA generation from laundry VOCs." Is this possible to do if the solvents are still limited to hydrocarbons? (For example, if the dry cleaning solvent is changed to supercritical CO2, of course, the pollution impact would be totally different.) 

Reviewer 2 Report

Overall: The authors measure the Volatile organic compounds (VOCs) from dry cleaning laundry machines using petroleum-based organic solvents. These can contribute to the ozone and SOA  formation. They also assessed photochemical ozone creation potentials and secondary organic aerosol potentials. It is an interesting and unique study of air quality potential due to dry cleaning machines, which many may not have thought about.

MAIN PAPER

Line 111. How was the flow rate determined for the volume of exhaust and chamber size? Also, are there any known/unknown reactions that can occur in the chamber before collection? Was this considered?

Line 116. When desorbing sorbent into GC-MS, when is quantification done with GC-FID as mentioned in on Line 108? Is there a split to each detector in same system? An FID system is not mentioned in this section.

Line 126. How are the three repeated VOC analyses done? Was it three separate collections of exhaust? What method is used, Relative Sensitivity Factors RSF? Also, is the qualitative measurement was a fourth collection? It is not clear from the text.

Figure 1. What are the concentration units? How one measure negative concentrations? Confusing; never saw analytical chemical data presented in this manner. There should be a better way to present this data.

May have missed it, but what estimated burden of these machines to total SOA.

SUPPLEMENTARY MATERIAL

Table S2. Is the 98% total of ozone precursors, or includes non-ozone forming as well? Is this the material used before the dry cleaning process started? Be clear.

Figure S2. Hard to read plot labeling. Increase font and bold.

Figure S1 & S2: S1 says drying exhaust sampled from 0-18min, while S2 says 40min. Why different. Explain.

Figure S2. Seems better to be in Main Paper. Figure 1 in Main Paper is confusing and makes no sense (+1.5 to -1.5?)

Round 2

Reviewer 1 Report

I feel that the improvements are rather limited. Thus, I do not recommend publication in its current form.